# Comparative Serum Proteome Analysis Indicates a Negative Correlation between a Higher Immune Level and Feed Efficiency in Pigs

**DOI:** 10.3390/vetsci10050338

**Published:** 2023-05-10

**Authors:** Siran Zhu, Jinglei Si, Huijie Zhang, Wenjing Qi, Guangjie Zhang, Xueyu Yan, Ye Huang, Mingwei Zhao, Yafen Guo, Jing Liang, Ganqiu Lan

**Affiliations:** 1College of Animal Science & Technology, Guangxi University, Nanning 530004, China; 1918401012@st.gxu.edu.cn (S.Z.); jinglei7139@126.com (J.S.);; 2Guangxi State Farms Yongxin Animal Husbandry Group Co., Ltd., Nanning 530004, China; 3Guangxi Botanical Garden of Medicinal Plants, Nanning 530023, China

**Keywords:** proteome, biomarkers, feed efficiency, serum, pig

## Abstract

**Simple Summary:**

In this study, based on the analysis of the feeding behavior of 350 Yorkshire pigs (90–160 days old), 24 pigs with a high- and low-feed rate (12 high and 12 low, half male and female, 80 days old) were selected for serum proteome sequencing. The results of the biological enrichment of differentially expressed proteins in the high-feed efficiency group and low-feed efficiency group were analyzed, and it was found that differentially expressed proteins were significantly enriched in metabolism-related pathways (glucose metabolism, lipid metabolism, immunity, and inflammation). The differentially expressed proteins were significantly enriched and significantly down-regulated in each immune pathway. Therefore, a higher level of immunity may not be conducive to improve the feed efficiency of pigs. The results of this study provide a favorable reference for the selection of early pig feed utilization efficiency in the actual production process.

**Abstract:**

Identifying and verifying appropriate biomarkers is instrumental in improving the prediction of early-stage pig production performance while reducing the cost of breeding and production. The main factor that affects the production cost and environmental protection cost of the pig industry is the feed efficiency of pigs. This study aimed to detect the differentially expressed proteins in the early blood index determination serum between high-feed efficiency and low-feed efficiency pigs and to provide a basis for further identification of biomarkers using the isobaric tandem mass tag and parallel reaction monitoring approach. In total, 350 (age, 90 ± 2 d; body weight, 41.20 ± 4.60 kg) purebred Yorkshire pigs were included in the study, and their serum samples were obtained during the early blood index determination. The pigs were then arranged based on their feed efficiency; 24 pigs with extreme phenotypes were grouped as high-feed efficiency and low-feed efficiency, with 12 pigs in each group. A total of 1364 proteins were found in the serum, and 137 of them showed differential expression between the groups with high- and low-feed efficiency, with 44 of them being upregulated and 93 being downregulated. PRM (parallel reaction monitoring) was used to verify 10 randomly chosen differentially expressed proteins. The proteins that were differentially expressed were shown to be involved in nine pathways, including the immune system, digestive system, human diseases, metabolism, cellular processing, and genetic information processing, according to the KEGG and GO analyses. Moreover, all of the proteins enriched in the immune system were downregulated in the high-feed efficiency pigs, suggesting that a higher immune level may not be conducive to improving feed efficiency in pigs. This study provides insights into the important feed efficiency proteins and pathways in pigs, promoting the further development of protein biomarkers for predicting and improving porcine feed efficiency.

## 1. Introduction

In pig production, feed costs account for over 60% of the total costs [1], which is the highest investment in the short lives of pigs. Therefore, improving feed efficiency (FE) is instrumental in reducing feed costs, increasing profits, and improving the environment [2]. The main solution to improve FE is genetic improvement [3]. Individual pigs within the same population often exhibit significant variation in the FE phenotype, regardless of their similar growth environment, genetic backgrounds, and management methods [4]. At present, two closely related indexes used to evaluate and predict FE traits are feed conversion rate (FCR) and residual feed intake (RFI) [5,6,7]. FCR is calculated as the amount of feed consumed per unit weight gain, while RFI is calculated by determining the discrepancy between individual animals’ feed intake as observed and as anticipated based on the average daily gain and backfat [8,9].

Although genome-wide association [10,11] and transcriptome analyses [12,13] have identified some important genes and pathways related to FE traits, they do not comprehensively explain the complex regulatory processes as they may be affected by posttranslational modifications [14], protein−protein interactions (PPIs) [15], and protein products [16]. Therefore, research at a protein level can better understand the molecular mechanism behind FE traits. Over recent years, several proteomic studies on pigs have been conducted, including reports on skeletal muscle [17,18,19], serum [20,21], heart [22], small intestine [23], liver [24], and spermatozoa [25]. Large-scale proteomic research is crucial to understanding the molecular mechanisms present within important economic traits in pig production, including FE. Several studies have identified FE-associated proteins and pathways in pigs. For example, three proteins—Gelsolin, Vitronectin, and SerpinA3—may have direct and significant biological involvement in the processes that regulate RFI and, ultimately, feed efficiency, through homeostasis and energy use [26]. Moreover, comparing the high-RFI and low-RFI pigs, the results of previous studies showed that the low-RFI pigs expressed more genes related to protein synthesis and less genes related to mitochondrial energy metabolism. Low-RFI pigs, as opposed to high-RFI pigs, had higher levels of two glycolysis-related proteins, fructose-biphosphate aldolase A and glyceraldehyde-3-phosphate dehydrogenase, while mitochondrial oxidative proteins, such as aconitase hydratase, ATP synthase subunit, and creatine kinase S-type, were less abundant [27,28]. For example, Fu [16] conducted a proteomic study on the skeletal muscle tissue of pigs with high-feed efficiency (HFE) and low-feed efficiency (LFE), and found a negative correlation between the mitochondrial metabolism of the skeletal muscle tissue and FE. By analyzing six commercial pig proteomes with extreme FE, another study found that 1219 proteins are expressed differently and Ras homolog family member A(RHOA), hematopoietic cell-specific Lyn substrate 1 (HCLS1), ezrin (EZR), cell division cycle 42 (CDC42), and Rac family small GTPase 1 (RAC1) may regulate FE in pigs according to the findings from comparing the proteomes of HFE and LFE pigs’ small intestine tissues [23]. However, the potential predictive efficacy of DEPs (differentially expressed protein (DEPs)) in young pigs on the important economic traits of grower-finishing pigs has not been fully clarified, for example whether the early selection of breeding stocks can expand the testing scale; increase selection intensity, especially for low-heritability traits (such as RFI); and achieve rapid genetic progress [28]. Consequently, an advanced proteomics analysis of serum proteins will make it easier for us to comprehend how FE traits and early blood index determination proteins are related and promote further identification of the biomarkers.

This study’s purpose was to investigate DEP in the serum of early performance tests of HFE and LFE pigs in a commercial environment. In the first step, the total proteins were separated from the sera of extreme HFE and LFE pigs calculated from the growth data using tandem quality tags (TMT). The DEPs between these two groups were exhaustively analyzed and randomly selected DEPs were validated using parallel reaction monitoring (PRM). Finally, the functions of the DEPs were analyzed using GO and KEGG enrichment analyses.

## 2. Materials and Methods

### 2.1. Animals and Blood Collection

The experimental design and protocol used in this research are shown in Figure 1. The purebred Yorkshire pigs (*n* = 350 (boar, 150; sow, 200)) that were used for this research were raised on a commercial farm. All of the pigs were weaned on day 21 and moved to the same feeding environmental fattening house (10 pigs per pen) on day 70. Each pig was marked with a distinct electrical tag on the ear throughout the testing phase, and the production performance was measured 10 days after adaptation (80 days of age). The pig performance test system (Kansas, America) of Osborne feed intake recording equipment (FIRE) was used to collect the growth data of the experimental pigs from day 90 to day 160, which was the fattening period for the pigs. For the derived data, the phenotypic data outliers were eliminated using Excel, then a regular distribution test and descriptive statistical analysis were carried out using SAS software (version SAS9.1), and a correlation analysis and significant difference test were carried out with the GLM module in SAS 9.1. Through the calculation of growth data, the daily feed intake (DFI), individual weight, average daily gain (ADG), and FCR data of the experimental pigs were obtained. When the pig’s weight reached 100 kg, the backfat thickness (BF) of the experimental pig was measured using a B-mode ultrasonic instrument. The equation used to predict the RFI was RFI = ADFI − [b1i × (onBW − 30) + b2i × (offBW − 100) + b3 × metamidBW + b4 × ADG1 + b5 × BF], the on-BW and the off-BW refer to measuring body weight at 90 and 160 days, and the metamid-BW = [(on-BW + off-BW)/2] × 0.75, which was previously described [29]. All of the pigs were given the same basic food and were allowed to eat and drink freely.

Blood samples were collected from each pig while fasting at 80 ± 1.15 days and they were coagulated at 20 °C for 30 min. The serum was separated by centrifugation at 2000 rpm for 15 min at 4 °C, conserved at −80 °C until use, and combined with a protease inhibitor. RFI and FCR indicators were used to measure the FE traits, and 24 pigs with extreme phenotypes were grouped as HFE and LFE, with 12 pigs in each group; thus, there were six boars and six sows in each group.

### 2.2. Protein Labeling and LC-MS/MS

The cell fragments of the serum samples were removed by centrifugation (12,000 rpm, 4 °C, 10 min), and the supernatant was moved to a new centrifuge tube. TMT-based quantitative analysis was used to perform proteomic analysis on the samples of the pig serum, as previously described [30]. The first 12 high-abundance proteins were removed using an abundance kit (Thermo Fisher, Waltham, MA, USA), and a BCA kit was used to quantify the protein concentration.

The protein was digested with 5 mM dithiothreitol (DTT) (30 min, 56 °C), alkylated with 11 mM iodoacetamide (15 min, 20 °C in the dark), and diluted by adding 100 mM triethylammonium bicarbonate to a urea concentration of ˂2 M. The peptides were desalted using a Strata X C18 solid-phase extraction column (Phenomenex) following trypsin digestion, and then vacuum dried. The peptide was reconstituted in 0.5 M triethylammonium bicarbonate and handled in accordance with the TMT kit’s manufacturer’s instructions. High-pH reverse-phase high-performance liquid chromatography (HPLC) was used to separate the tryptic peptides using an Agilent 300Extend C18 column (5 m particles, 4.6 mm diameter, and 250 mm length).

### 2.3. Database Search and Bioinformatics

Secondary mass spectrometry data were retrieved using the MaxQuant software (version v1.5.2.8). The search parameter was set to Sus_scrofa_9823_PR, and an anti-database was added to compute the false positive frequency (FDR) as a result of random comparison. To determine how contaminated proteins affected the outcomes of the elimination process, a standard pollutant library was added to the database. Trypsin/P is called lyase, and up to two parts were allowed for each deletion.

### 2.4. Bioinformatics Analyses

The phenotypic statistical analysis and correlation of RFI, FCR, average daily feed intake (ADFI), ADG, 100 kg BF, On-BW (90-day BW), Off-BW (160-day BW), and Metamid-BW in the Yorkshire pig population were conducted using SAS software. InterProScan and KAAS were used to perform GO and KEGG annotations. The Perl module was used to perform an enrichment analysis of GO terms and KEGG pathways. With a corrected *p* < 0.05, the GO and pathways were deemed significant. A two-tailed Fisher’s exact test was employed to examine the enrichment of the DEPs versus all of the identified proteins. The STRING database version 10.1 was used to retrieve all DEPs and draw PPIs (protein–protein interaction network). The STRING interaction network was visualized using Cytoscape.

### 2.5. PRM-MS Analysis

To confirm the accuracy of the protein expression trend discovered by TMT analysis, 10 proteins with significant differences in TMT results were detected through parallel reaction mass spectrometry (PRM-MS) by post-translational modification (PTM), and the related proteins were quantitatively expressed (Hangzhou, China). TMT data were used to identify identifiable peptides for the target proteins, and two or more distinct peptide sequences were chosen for PRM analysis. The sample was processed according to the TMT analysis scheme. First, the protein was extracted, then the high abundance protein in the sample was removed using the kit, and finally, trypsin digestion was carried out. 

## 3. Results

### 3.1. Basic Statistics of Porcine Performance and FE

The average RFI of 24 Yorkshire purebred pigs was 0.02 ± 0.31 (Mean ± SD, kg) and the FCR value was 2.55 ± 0.37 (mean ± SD). There were significant differences in the phenotypic characteristics of RFI and FCR between the HFE group and LFE group (*p* < 0.001) (Table 1). In comparison with the LFE group, the HFE group’s phenotypic value of ADFI was lower (*p* < 0.001), and the values of RFI and FCR in the HFE group were also lower than those in LFE (*p* < 0.001, *p* < 0.001). In addition, other related traits between the HFE and LFE groups, including ADG, on-BW, off-BW, 100 kg BF, and Metamid-BW, were compared; however, there were no significant changes (*p* > 0.05) (Table 1).

Moreover, the correlation analysis revealed that the phenotypic characteristics of RFI and FCR had a significantly positive correlation (R = 0.79, *p* < 0.0001), and ADFI had a significantly positive correlation with RFI and FCR (R = 0.56, *p* < 0.0001; R = 0.29, *p* < 0.0001, respectively) (Appendix A). RFI was not correlated with other phenotypes such as ADG, 100 kg BF, and BW (*p* > 0.05). However, FCR had a significantly negative correlation with ADG (R = −0.44, *p* < 0.01) and a significantly positive correlation with on-BW (R = 0.35, *p* < 0.0001). The growth data of 350 pigs were collected and analyzed according to the reported calculation equations for RFI, FCR, ADFI, ADG, 100 kg BF, On-BW, Off-BW, and Metamid-BW. The results are shown in Appendix A.

### 3.2. Proteomic Differences between the HFE and LFE Groups

Overall, 468,379 secondary spectra were obtained from liquid chromatography−mass spectroscopy (LC/MS) analysis. In total, 1460 proteins were detected from 9091 unique peptides, and 1362 quantifiable proteins were identified by quantitative proteomic analysis (FDR < 1%) (Figure 2A). The maximum length of the peptide sequenced by mass spectrometry was 9 amino acids, which generally ranges from 7 to 37 amino acids (Figure 2B). The range of proteins identified by mass spectrometry was 0–20 kD, 20–40 kD, 40–60 kD, 60–80 kD, and 80–100 kD. The number of proteins with molecular weights accounted for 83.7% (Figure 2C). Furthermore, 77.88% of the 1460 identified proteins contained more than two peptides (Figure 2D).

Principal component analysis (PCA) performed to evaluate the relative expression levels of each protein revealed that the degree of protein expression revealed the repeatability of the biological processes (Figure 3A). Among the 1460 proteins identified, the expression of 137 was significantly different between the HFE and the LFE groups (fold change > 1.2; *p* < 0.05); 44 proteins were upregulated and 93 were downregulated in the LFE group compared with the HFE group (Figure 3B). In Appendix A, details of the identified DEPs are presented, along with protein names and UniProt accession codes. Details of the identified DEPs, including protein names and UniProt accession numbers, are listed in Appendix A.

### 3.3. GO Annotation and KEGG Enrichment of DEPs

Using the UniProt-GOA database and InterProScan, GO annotation analysis was performed on all DEPs. According to the GO analysis, the DEPs were annotated to cellular components, biological processes, and molecular function terms. The action pathways of 30 genes were significantly enriched (*p* < 0.05), including 14 biological functions terms, 8 cellular components terms, and 8 molecular functions terms. The outcomes of the study of the cellular components showed that most DEPs were located in the cis-Golgi network, proteasome regulatory particles, proteasome accessory complexes, and ficolin-1-rich granule membranes (Figure 4A). Moreover, the biological process term analysis revealed that DEPs mainly participated in biological processes, such as the ubiquitin- and modification-dependent protein catabolic processes, Fc-gamma receptor signaling pathway process, and nucleic-acid-templated transcription process (Figure 4B). Similarly, the molecular function analysis’s findings suggested that most DEPs were enriched in translation elongation factor activity, lipoprotein particle receptor binding, translation factor activity, and RNA binding (Figure 4C).

A functional enrichment study using the KEGG database was carried out to look further into DEP-related biological processes. The results revealed that DEPs in the HFE and LFE groups were significantly involved in nine KEGG pathways (*p* < 0.05), such as the type II diabetes mellitus, c-type lectin receptor, and mRNA surveillance signaling pathways (Figure 5). The GO and KEGG enrichment analysis results of differentially expressed proteins are shown in Appendix A.

### 3.4. PPIs Network Construction and Analysis

A considerable subnetwork with 84 nodes and 157 edges was discovered through network analysis of the DEPs (Figure 6). Twenty-three hubs with a minimum betweenness score of 100 made up this subnetwork, each of which interacted with at least four other nodes. In this subnetwork, 23 hubs interacted with a minimum of four other nodes and had a minimum betweenness score of ˃100. The top three among the 23 hubs were alpha 2-HS glycoprotein (AHSG), protein phosphatase 2 scaffold subunit alpha (PPP2R1A), and tubulin beta class I (TUBB). 

### 3.5. DEPs Validated by PRM

To validate the DEPs discovered through the TMT analysis, a PRM test was carried out. Given that the PRM assay required the unique peptide of the proteins, proteins with unique peptides were randomly selected for analysis. Ten DEPs (upregulated: AHSG, serpin family G member 1 (SERPING1), and complement factor I (CFI); downregulated: creatine kinase B (CKB), angiopoietin-like 8 (ANGPTL8), prolyl 4-hydroxylase subunit beta (P4HB), peroxiredoxin 2 (PRDX2), pentraxin 3 (PTX3), proprotein convertase subtilisin/kexin type 9 (PCSK9), and leukocyte cell-derived chemotaxis 2 (LECT2)) were selected for PRM assay. The PRM results revealed that the expression values of AHSG, SERPING1, and CFI were higher in the HFE group than in the LFE group, but without significant differences (*p* > 0.05). The CKB, ANGPTL8, P4HB, PTX3, PRDX2, PCSK9, and LECT2 expression levels were all significantly lower in the HFE group compared with the LFE group (*p* < 0.05). The fold changes for these proteins using the PRM assay showed the same trend as the TMT results between the HFE and LFE groups (Table 2).

## 4. Discussion

In this research, the serum proteins that were differently expressed in the early blood index determination between HFE and LFE were compared using TMT profiles, and 137 differentially abundant serum proteins were identified. Simultaneously, several important signaling pathways associated with FE were detected. Most DEPs enriched in immune-related pathways were significantly downregulated in the HFE group, indicating that HFE pigs may have a less efficient immune system. Based on the functional analysis of differentially expressed proteins in the important immune-related pathways, some proteins may be potential candidates for affecting porcine FE, and the mechanism of their function affecting FE needs further investigation.

Measuring the full-range production performance of large groups in China is very difficult because of the large, relatively scattered, and independent population of breeding pigs. Therefore, early prediction of production performance has become an urgent need for the pig breeding industry. While there have been considerable advancements in animal genetics and management, feed costs still account for roughly 60% of the production costs in the swine industry. No genome or biomarker has been identified for FE, especially in early blood index determination. Breeders, using the method of combining animal genetics theory with practice, have made great contributions to the breeding of important economic traits in breeder animals, and at the same time actively improve the production management program to provide a good growth environment for breeder livestock. These inputs can appropriately save production costs, but cannot solve the fundamental problems. Improving pig feed efficiency will solve a fundamental problem because feed efficiency accounts for 60% of the production cost. There is a close relationship between the protein metabolism pathway and feed efficiency in animals. The protein metabolism pathway refers to the process of digestion, absorption, transport, synthesis, and degradation of protein in animals, while feed efficiency refers to the ratio of productivity or weight gain to feed intake. The impact of the protein metabolism pathway on feed efficiency can be summarized as follows. Digestion and absorption efficiency: factors such as digestive enzyme activity, intestinal health status, and amino acid supply can affect the digestion and absorption efficiency of protein, and thus affect feed utilization and feed efficiency. Protein synthesis: protein synthesis requires multiple amino acids and other nutrients. If the content of certain essential amino acids in the feed is insufficient, it will limit protein synthesis and productivity, thereby reducing feed efficiency. Physiological state: the demand for protein in an animal and its utilization efficiency may vary under different physiological conditions, which also affects feed efficiency. Therefore, understanding the protein metabolism pathway is crucial for optimizing feed formulation and improving feed efficiency.

The genome and biomarker of FE have not been determined, especially in early performance testing. If FE biomarkers that can be identified in the early performance testing stage can be found, the production cost will be greatly reduced. As the source of biomarker detection, blood is the most ideal biological fluid. When testing, taking samples can minimize the damage to pigs as much as possible. With the development of high-throughput sequencing technology, mining new physiological markers of FE has become a feasible scheme. RFI and FCR are generally used to evaluate FE traits because of their high genetic and phenotypic correlation; previous studies have shown that the genetic and phenotypic correlations between RFI and FCR range from 0.67 to 0.89 [31,32,33]. Similar to previous studies, the phenotypic correlation in this study between RFI and FCR was 0.79, using the same assessment method.

In order to improve the FE of pigs, we should not only improve the feed digestion, metabolism, and living environment, but also pay attention to heat stress and inflammation. Previous studies have shown that immune challenges in pigs are important factors that affect FE. Additionally, HFE pigs have been suggested to have a greater inhibitory ability on inflammatory and immune responses than LFE pigs [34]. In the early stage of our laboratory, we used the experimental samples (*n* = 350) and the growth data (including feed efficiency, daily gain, average feed intake, etc.) to analyze the correlation between biochemical indexes and growth data. It was found that the correlation coefficient between TG (Triacylglycerol) and Glu (Glucose) in blood was the highest (r = 0.65, *p* < 0.0001). The second was the correlation coefficient between HDL-C (High density liptein cholesterol) and Glu, TG, and LDL-C (Low Density Lipoprotein) (r = 0.51~0.59, *p* < 0.0001). TG was negatively correlated with body length and height, and positively correlated with FCR and corrected 30–100 kg FCR. LDH (Lactate dehydrogenase) was negatively correlated with FCR, corrected 30–100 kg FCR, and RFI. sCr (Serum creatinine) was negatively correlated with RFI, ADFI, FCR, corrected 30–100 kg FCR, off-BF, and corrected 100 kg BF (r = −0.13~−0.25, *p* < 0.05), and was positively correlated with eye muscle area (r = 0.13, *p* < 0.05). CRP (C-reactive protein) was significantly negatively correlated with off-eye muscle area, corrected eye muscle area by 100 kg, and corrected eye muscle thickness by 100 kg (r = −0.21~−0.23, *p* < 0.01), and positively correlated with FCR (r = 0.13, *p* < 0.01). Glu and IL-6 were positively correlated with FCR and corrected 30–100 kg FCR (r = 0.12~0.15, *p* < 0.05), and IL-6 was positively correlated with RFI and 100 kg backfat thickness EBV (r = 0.18~0.21, *p* < 0.05). At the same time, the results of TNF-α, IL-6, and CRP in the H/LFE group were analyzed, and it was found that there was no significant difference in these three immune indexes between the two groups. Therefore, there is a certain relationship between feed efficiency and immunity or inflammation.

Challenges still remain regarding the improvement of FE for pigs, such as the ability to digest, efficient metabolism, reduced maintenance or activity requirements, heat stress, and inflammation. Previous studies have shown that immune challenges in pigs are important factors affecting FE, such as C-type lectin receptors and chemokine signaling pathways, and HFE pigs may have a higher ability to inhibit inflammation and immune response than the LFE pigs [34]. The results of the GO and KEGG enrichment analysis of the differential proteins in the high and low FE groups revealed that LSP1, PTPN11, CXCL10, STAT5B, RHOA, and PRKCD were significantly downregulated in the HFE group and enriched in the C-type lectin receptor and chemokine signaling pathways. CXCL10 was also involved in the TNF-signaling pathway. Therefore, we speculate that LSP1, PTPN11, CXCL10, STAT5B, RHOA, and PRKCD may be important candidate proteins affecting FE.

Protein tyrosine phosphatase non-receptor type 11 (PTPN11) is a crucial protein that may influence the FE characteristic, which encodes tyrosine phosphatase (SHP-2) and has a crucial function in signaling events downstream of the JAK/STAT signaling pathway [35]. It has been shown that the expression of SHP-2 affects the metabolism of fat and glucose in mice, as well as energy balance [36]. Additionally, many cytokines and growth factors, including interleukin (IL)-6, growth hormone, epidermal growth factor, platelet-derived factor, and interferon, are transmitted through the JAK/STAT signaling pathway [37,38,39]. The results of this experiment showed that the expression of PTPN11 was significantly downregulated in the HFE group, and it was speculated that the expression of SHP-2 changed and affected the expression of PTPN11. STAT5B is another protein that may affect FE and plays an important role in body growth, hematopoietic function, and immune response in organisms. A recent human study showed that a STAT5B gene mutation resulted in reduced serum IGF1 in most patients with immunodeficiency growth hormone insensitivity, accompanied by growth retardation and poor growth hormone therapy [40]. The plasma concentration of IGF-1 and insulin in low-RFI cattle were higher than that in high-RFI Nellore cattle, but the urea concentrations were lower than those in high-RFI Nellore cattle. The results indicated that for Nellore cattle, plasma concentrations of urea, IGF-1, and insulin can be employed as markers of FE [41]. Therefore, we speculate that the difference in the expression of STAT5B in different FE groups will affect the content of IGF-1 in the serum, and thus affect the FE of pigs. RAC2 is a member of the Rho family of GTPases, which plays an important role in cells and participates in the regulation of various signaling pathways, such as cell cycle regulation, apoptosis, and immune response. The deletion of RAC2 in mice also induced hypogammaglobulinemia (IgM and IgA), accompanied by reduced IgM AFC (Ab-forming cells) secretion, and thus impaired the humoral immune response. This injury did not appear to be related to an intrinsic block of B cell differentiation, but to a defected in BCR-mediated RAC2-dependent signaling. The decrease in RAC2 calcium flux proved that BCR signal transduction damage and BCR antigen binding decreased in RAC2 (-/-) mice [42,43]. In inflammatory cells, RAC2 is a crucial signal transduction factor that is essential for the formation of reactive oxygen species (ROS) in reaction to agonists of receptors, such as growth factors or inflammatory cytokines [44]. Further analyses were performed by enrichment and interaction analysis using DEGs and GSEA methods to detect changes between native chickens in different RFI groups. Researchers have shown a connection between feed efficiency and the genes involved in immunological response, mitochondrial function, stimulus response, and inflammatory response. Among them, RAC 2 is one of the key genes affecting feed utilization efficiency in native chickens, which may affect FE by participating in the production of ROS and inflammatory reaction [45]. This provides strong evidence for the results of this experiment. C-X-C motif chemokine ligand 10 (CXCL10) is crucial in attracting inflammatory cells to areas of tissue injury [46]. Lymphocyte-specific protein 1 (LSP1) has been identified as a regulator of neutrophil chemotaxis during inflammation and has a complex function in regulating leukocyte recruitment to inflamed sites. Protein kinase C delta (PRKCD) belongs to a family of serine/threonine kinases that encode protein kinase C δ (PKCδ) [47]. According to earlier research, the PKC protein is crucial for B cell signaling; autoimmunity; and control of the proliferation, death, and cell-type differentiation [48,49]. RHOA is a key protein involved in smooth muscle contractions. Recent studies have demonstrated that IL-4 can upregulate RHOA by promoting its transcription in human bronchial smooth muscle cells (hBSMCs) [50]. Therefore, the abundance of proteins related to immunity or inflammation was higher in the LFE group, suggesting that the immune efficiency of the HFE group may be lower than that of the LFE group.

Moreover, the network interactions of proteins showed that PPP2R1A, PSMC1, PSMA7, PSMC3, TUBB2A, and PSMD3 were closely related. These proteins participated in immune, metabolic, and genetic information processes, indicating the possible reasons for their effect on FE. 

## 5. Conclusions

In summary, the proteome of pig serum in the HFE group and LFE group was compared and analyzed in this study. In total, 1460 serum proteins of early-stage production performance measurement were identified, and 137 DEPs between the HFE and LFE groups were detected. All DEPs enriched in immune-related pathways were significantly downregulated in HFE pigs. Therefore, HFE pigs may have a less efficient immune system. These findings offer a new angle of view for examining potential genes that influence FE characteristics during early blood index determination. However, the precise relationship between these proteins and FE requires further elucidation; these putative candidate proteins should be further validated as relevant biomarkers of FE.

## Figures and Tables

**Figure 1 vetsci-10-00338-f001:**
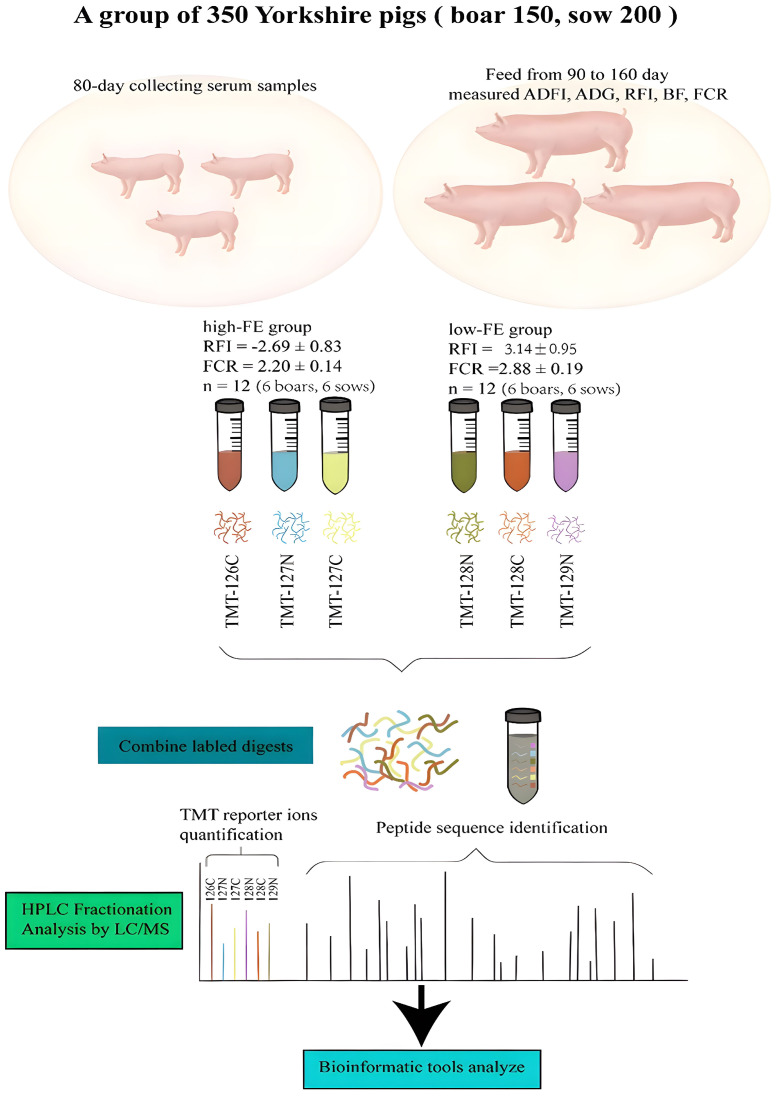
Overview of the experimental design used in this study.

**Figure 2 vetsci-10-00338-f002:**
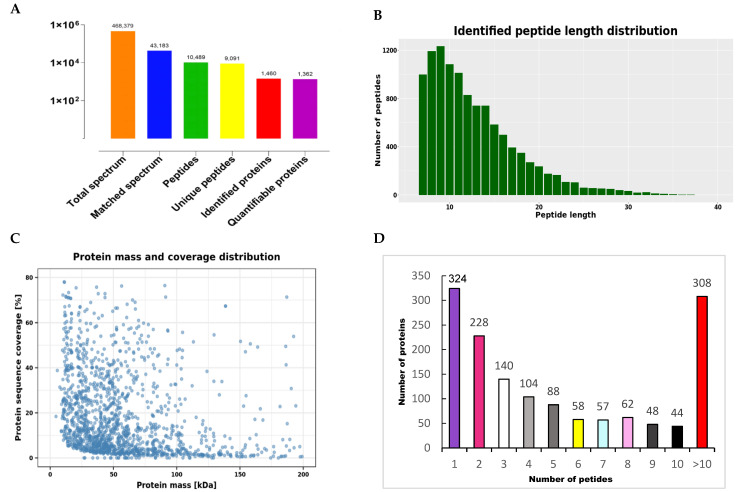
Protein identification and analysis using TMT. (**A**) Total spectrum, peptides, and proteins identified using the TMT method. (**B**) Identified peptide length distribution. (**C**) Protein mass and coverage distribution. (**D**) Protein distribution based on peptide number.

**Figure 3 vetsci-10-00338-f003:**
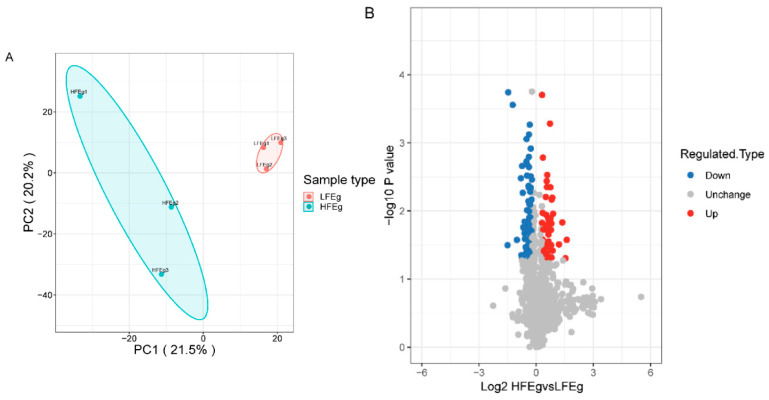
The identification of differentially expressed proteins between HFE and LFE groups using TMT technology. (**A**) Based on the principal component analysis of the proteins identified in this study. The orange and azure dots represent LFE and HFE groups, respectively. (**B**) Volcano plot of all proteins identified by TMT. The red and blue dots represent upregulated and downregulated proteins under the screening condition, respectively. Grey dots represent unchanged proteins.

**Figure 4 vetsci-10-00338-f004:**
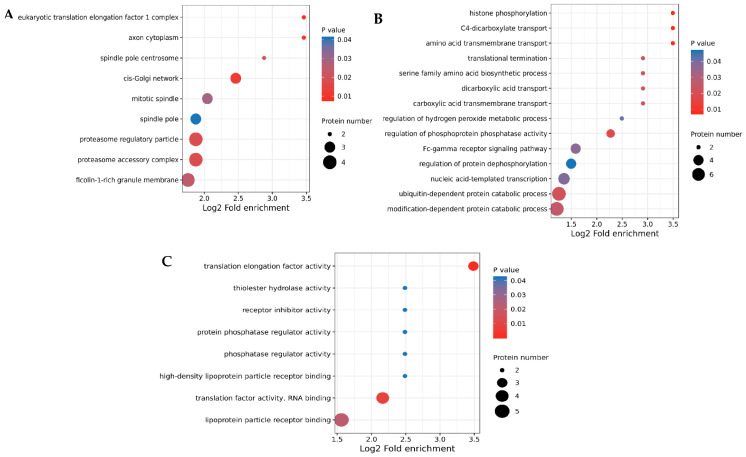
Gene ontology classification enrichment analysis of all DEPs. GO terms for the cellular component (**A**), biological process (**B**), and molecular function (**C**). The *X* axis and *Y* axis are the mean enrichment factors and GO terms, respectively. The circle size indicates the amount of protein, and the different colors indicate different *p*-values. The GO enrichment analysis was tested using Fisher’s exact test, and *p* < 0.05 is considered significant.

**Figure 5 vetsci-10-00338-f005:**
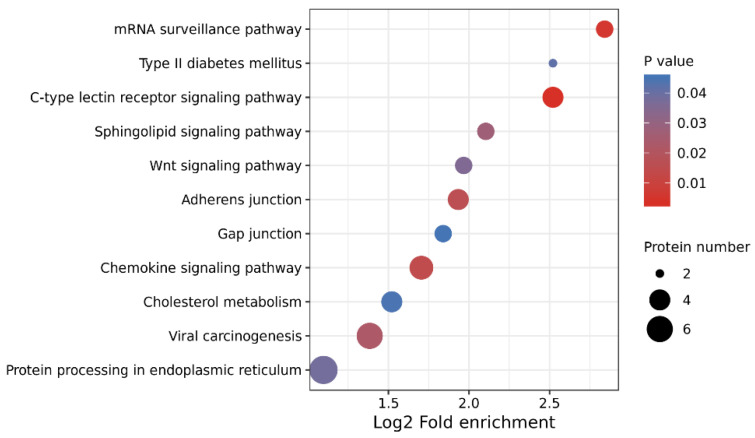
Kyoto Encyclopedia of Genes and Genomes (KEGG) pathway enrichment analysis of the DEPs. The *X* axis and *Y* axis are the mean enrichment factors and KEGG pathways, respectively. The circle size indicates the amount of protein, and the different colors indicate different *p*-values. The KEGG enrichment analysis was tested using Fisher’s exact test, and *p* < 0.05 is considered significant.

**Figure 6 vetsci-10-00338-f006:**
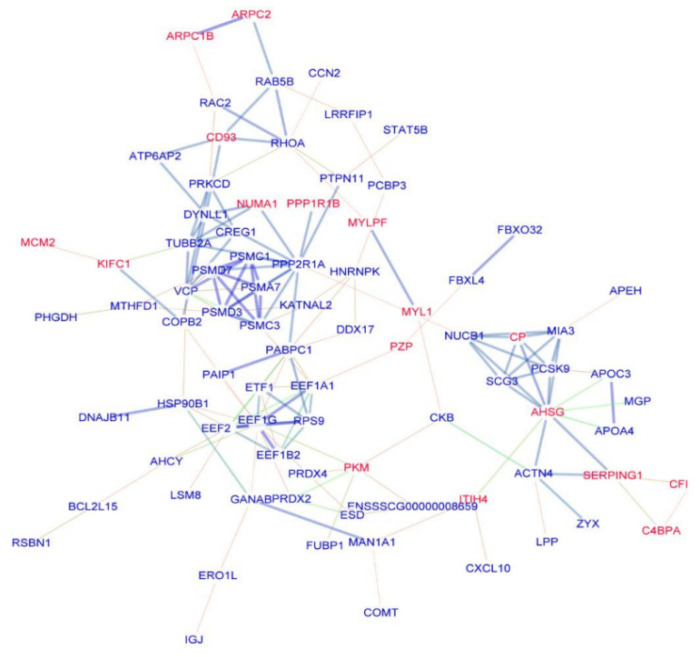
Interaction network analysis of the DEPs. The red and blue represent upregulated and downregulated proteins, respectively. Network nodes and edges represent proteins and protein-protein interaction, respectively.

**Table 1 vetsci-10-00338-t001:** Phenotypic value description of FE and related traits.

Parameter	LFE (Mean ± SD)	HFE (Mean ±SD)	*p*-Value ^a^
N	12	12	
RFI (kg)	3.14 ± 0.95	−2.69 ± 0.83	<0.001 ***
FCR	2.88 ± 0.19	2.20 ± 0.14	<0.001 ***
ADFI (kg/d)	2.90 ± 0.20	2.31 ± 0.18	<0.001 ***
ADG (kg/d)	1.01 ± 0.10	1.03 ± 0.07	0.56
100 kgBF (mm)	13.05 ± 3.35	12.46 ± 2.41	0.62
On-BW (kg)	41.67 ± 5.46	40.33 ± 6.70	0.46
Off-BW (kg)	112.37 ± 8.66	110.57 ± 5.92	0.56
Metamid-BW	57.76 ± 4.05	56.59 ± 2.99	0.43

The data are expressed as the mean values ± standard deviation (SD). RFI, residual feed intake; FCR, feed conversion ratio; ADFI, average daily feed intake; ADG, average daily gain; 100 kg BF, 100 kg back fat thickness; On-BW, 90-day body weight; Off-BW, 160-day body weight; Metamid-BW, intermediary metabolism body weight. ^a^, *p*-values were determined using one-way analysis of variance test (*** *p* < 0.001).

**Table 2 vetsci-10-00338-t002:** Confirmation of DEPs detected in the TMT analysis using PRM analysis.

Protein Accession	Protein Gene	Fold Change (HFE/LFE) in TMT	*p*-Value in TMT	Fold Change (HFE/LFE) in PRM	*p*-Value in PRM
F1SFI7	AHSG	2.52	1.50 × 10^−2^	2.23	1.13 × 10^−1^
F1SJW8	SERPING1	1.65	1.30 × 10^−2^	1.26	1.24 × 10^−1^
F1S133	CFI	2.23	3.10 × 10^−2^	2.17	1.50 × 10^−1^
Q29594	CKB	0.83	3.50 × 10^−3^	0.39	4.48 × 10^−3^
T1UNN8	ANGPTL8	0.75	4.60 × 10^−3^	0.37	3.53 × 10^−3^
G9F6X8	P4HB	0.78	2.30 × 10^−2^	0.43	2.60 × 10^−3^
A0A287BAZ6	PRDX2	0.78	8.10 × 10^−3^	0.44	2.93 × 10^−3^
A0A287BKS2	PTX3	0.77	9.60 × 10^−3^	0.37	4.27× 10^−2^
I3LGB2	PCSK9	0.82	1.80 × 10^−2^	0.49	2.44 × 10^−4^
F1RHA9	LECT2	0.78	2.20 × 10^−2^	0.50	1.26 × 10^−2^

## Data Availability

The original contributions presented in the study are deposited in the iProX database under accession code ID: PXD038186 (http://proteomecentral.proteomexchange.org/cgi/GetDataset?ID=PXD038186 (accessed on 16 November 2022), further inquiries can be directed to the corresponding author.

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
