# Peer review of "Comparative Serum Proteome Analysis Indicates a Negative Correlation between a Higher Immune Level and Feed Efficiency in Pigs"

_vetsci, 2023, doi:10.3390/vetsci10050338_

Round 1

Reviewer 1 Report

Dear authors,

Please, find all comments, suggestions, and questions (please answer them) in the PDF file enclosed.

The present study contain an interesting dataset. Some precision is needed in the description of some terms and methods. Attention should be paid for editing error, there are many. The quality of the images must be improved, in some cases it is barely possible to read their content. The authors must review the coherence between their title, objectives (abstract and introduction), and conclusions. The title points to the interaction between feed efficiency and immune status, the abstract objectives and conclusion is on the importance of identifying candidate marker for feed efficiency, the introduction objective does not indicate if the study will evaluate candidate markers or the immune system, and the conclusion summarized both subjects. But my concern with this manuscript is the discussion section. The authors were not able to clearly describe why the indicate proteins may be candidates of feed efficiency markers. Additionally, throughout the discussion section, the authors simply described the function of some proteins on the immune system without explaining how they work together to impact feed efficiency. The authors appears to believe it is through greater energy consumption by the immune system, but they were not able to clearly describe the possible mechanisms behind it. 

Thank you

Author Response

Response to Reviewer 1 Comments

Point 1:What do the authors mean by early-stage performance test?

Response 1:I'm sorry for causing confusion with my improper wording. The performance measurement mentioned in the text refers to the entire pig fattening period (80-160d), and the early performance measurement mentioned refers to the blood collected before measurement (80±2d) and growth data at the time of measurement. By analyzing the growth data during the measurement period, pigs with extremely high or low feed efficiency were selected, and their collected blood was analyzed, which I defined as early performance measurement. The text has been modified to "early blood index determination". Through the analysis of marker genes related to growth rate, feed efficiency, and reproductive ability, rapid breeding can be achieved through blood testing. Effective screening before pig fattening can help us eliminate pigs with slow growth or poor growth, thereby improving breeding efficiency and economic benefits. At the same time, it can also provide reference for pig feed formula and feeding management, so that we can better grasp the growth of pigs and carry out feeding and management more effectively.

Point 2:Please, indicate average age and weight

Response 2:The information of age and weight has been added to the revised manuscript. 350 (age, 90±2d, body weight, 41.20 ±4.60 kg).

Point 3:The authors make no previous link between feed efficiency and environmental issues to get to this affirmation here. Please, clearly show the link.

Response 3: Relevant literatures have been cited. Reference [2] indicates that the environment of pigs with L/HFE was evaluated for acidification potential, freshwater eutrophication potential, and water resources. It was found that improving feed efficiency can reduce environmental harm, and HFE pigs have a 7% lower environmental impact than LFE pigs.

Point 4:This is an important one, but not the only. What about feed quality, use of additives?

Response 4:The words have been changed. Change "The keys to improving FE are genetic improvements" to "The main solution to improve FE is genetic improvement".

The study in reference [3] found that the use of improved feeding sequence during weaning could improve the feed efficiency of piglets and effectively reduce the weaning stress of piglets, but the effect of this improvement was not significant during the fattening period.

In the actual production, the best way to improve pig feed efficiency should be to comprehensively consider a variety of factors, including genetics, feed and additives, this paper is mainly discussed from the perspective of genetic improvement.

Point 5:Was it reported by other authors or are the authors explaining what was done in the present study? This sentence is confusing. Please, re-work the sentence.

Response 5:Thank you for your comments. The relevant sentences have been revised in the original text. For example, Fu [16] conducted a proteomic study on the skeletal muscle tissue of pigs with high-feed efficiency (HFE) and low-feed efficiency (LFE) and found a negative correlation between the mitochondrial metabolism of the skeletal muscle tissue and FE.

Point 6:Do the authors mean "pigs in early growing phase expressing HFE and LFE"?

Response 6:The grouping of HFE and LFE was obtained by calculating the growth traits after the performance test (age,160d) and selecting the extreme values for grouping.

The purpose of this experiment is to find the serum of pigs in L/HFE group at the beginning of determination (age, 80 ±5d) according to the grouping after the determination is over. Through the analysis of serum protein groups in this period, the early proteins of pigs related to feed efficiency are excavated. Early stage means the sampling time is early.

Point 7:This is not clear. After reading the whole manuscript and studying Figure 1, my understanding is that the experiment started when pigs had 70 days of age. After an adaptation period of 10 days (when pigs had 80 days of age) the only procedure that was performed as the blood sampling. Performance was assessed from 90 to 160 days of age. What do the authors mean by production performance here?

Response 7:This production performance test uses the Osborne automatic feeding system test column in the United States to identify and record the daily feeding times, each feeding time, each feeding intake and the weight each time entering the measuring column for the pigs in the same column (15 pigs, each pig has a corresponding electronic ear tag). The collected data are transmitted to the central control room and processed by the system.

At the same time, the backfat thickness of pigs at about 100kg was measured, which is also the main index for calculating pig feed efficiency.

Point 8:Please, reword. This sentence begins with "According to previous reports" and ends with "has been previously described.

Response 8:Make changes in the submitted manuscript.

Point 9:Review er:Please, provide a better image quality.

Response 9:All figures have been resubmitted to the original images with high quality.

Point 10:In the discussion part, the number of serum protein group LFE and HFE pigs

Response 10:The number of serum protein group LFE and HFE groups has been described in the materials and methods, in which LFE and HFE groups have 12 pigs in each group, half male and half female.

Point 11:These results have to be interpreted with care. The authors interpretation may lead to the assumption that immune suppressed individuals perform better whereas immune competent individuals perform worse. Considering that in this study animals were selected based on feed efficiency and not on "immune efficiency", and that animals were healthy (immune system not highly activated), I believe the interpretation must be done the other way around: HFE pigs have a less efficient immune system. In fact, Audet et al. (2015; doi: 10.2527/jas.2014-7872) reported that ADG was positively correlated with homocysteine levels (metabolite detrimental to pigs health) and homocysteine inhibits immune response. They suggested that piglets with higher growth rates have a more intense metabolism, generating more homocysteine and, as a consequence, impairing their immune system. The interpretation of those authors appears to be applicable to the present study, and may actually help explaining the present results.

Response 11:Thank you very much for your suggestion We have put "Therefore, a higher immune level may not be conducive to increasing FE in pigs." modified to "Therefore, HFE pigs may have a less efficient immune system."

Point 12:This sentence is not clear. Please. re-work.

Response 12:Thank you very much for your question, which I have revised in the submitted paper. The modifications are as follows: In order to improve the FE of pigs, we should not only improve the feed digestion, metabolism and living environment, but also pay attention to heat stress and inflammation.

Point 13:The link between the present results and the discussion that follows is missing. In the current format, the reader does not understand why the authors are discussing JAK/STAT and other pathways.

I suggest the authors to add a sentence indicating the present results that are related to the discussion that follows. For example, they could use the sentence at L353-357 "The study results revealed that immune signaling pathways were significantly affected by FE, such as the c-type lectin receptor and chemokine signaling pathways. LSP1, PTPN11, CXCL10, STAT5B, RHOA, and PRKCD were all significantly downregulated in the HFE group and enriched in the c-type lectin receptor and chemokine signaling pathways. "

It would be interesting to indicate which of those are part of the JAK-STAT pathway, the RAC pathway, etc. So readers will clearly understand the link.

Response 13:Thank you very much for your comments. I have adjusted the sentence order from L353-1357 to L288. After the adjustment, the meaning of the article is expressed more smoothly. Thank you again for your comments.

Point 14:The simply description of some proteins' functions is not enough to explain the results. Besides the sentence "The weight and energy intake of mice are impacted by SHP-2 expression. Reduced insulin-induced inhibition of hepatic glucose production" there is no real indication of how the indicated proteins or pathways impact the interpaly between immune system and feed efficiency.

Based on the present results, how do these protein act and/or interact to impact the immune system in a negative manner in HFE?

Response 14:Thank you very much for the teacher's question, I have added an introduction to the relevant pathways and the relationship between immunity and metabolism.

The content added is as follows: These proteins are selected because these proteins are involved in the same signaling pathways at the same time. CXCL10 participates in TNF signaling pathway, CXCL10, STAT5B, RHOA, PRKCD, LSP1 were all significantly downregulated in the HFE group C-type lectin receptor signaling pathway, and Chemokine signaling pathway. These are immune-related signaling pathways. According to the previous results of our laboratory: CRP\ IL-6\ TNF- α is positively correlated with FCR, and IL-6 is also positively correlated with RFI. Therefore, the immunity of the body will have an impact on feed efficiency. The study of this paper has some limitations, which can only show that these proteins are potential targets that affect feed efficiency. As for the mechanism of action, we need to continue to study further.

Point 15:What is the importance for the present study? If it is not essential, this sentence could be removed.

Response 15:The relevant description has been deleted in the article.

Point 16:What is the importance for the present study? If it is not essential, this sentence could be removed.

Response 16:This sentence is my description of the function of the STAT5B gene after consulting the literature, indicating why the gene chose this gene for discussion, so I think it can be retained.

Point 17:But the present study is about pigs. What is the relevance of this information?

Response 17:This is to quote the contents of other people's articles to illustrate the correlation between feed efficiency and immunity. I have revised the expression in the article.

Point 18:The fact that these proteins are important to the immune system and they are differentially expressed in the present study, does not make them appropriate candidates for selection aiming feed efficiency unless the authors are able to clearly explain how they interfere with feed efficiency.

The simply description of some proteins' functions is not enough to explain the results. How and why these proteins may be candidate genes? What are the most probable metabolic events leading them to interfere with feed efficiency?

Response 18:Thank you very much for asking this question.

Feed efficiency is a complex character comprehensively reflected in a variety of metabolic processes of the body. because of its importance in production, there have been a large number of studies on the mechanism of feed efficiency.

For example, muscle growth and fat deposition in the body have been found to be associated with it. However, the regulation mechanism that determines feed efficiency is still difficult to be clear, and all the conclusions obtained are possible, especially the study of RFI indicators.

According to the study of this paper, these proteins are potential candidates related to pig feed efficiency.

In this paper, based on proteomic screening and difference analysis, it is found that there are significant differences in these proteins between high and low groups, and verified by PRM, of course, these results are only based on correlation analysis, and there is no causal mechanism research.

Therefore, we modify the conclusion that LSP1, PTPN11, CXCL10, STAT5B, RHOA, and PRKCD proteins are potential candidate markers associated with FE, and the function of related proteins needs further experimental verification.

Point 19:I do agree with this general conclusion but the current discussion does not allow such conclusion because the authors did not demonstrate basically any interaction between the energy metabolism and the immune system.

The present KEGG analysis point to C-type lectin receptor signaling (related to immune response) and Cholesterol metabolism (related to energy metabolism). Is there any metabolic relationship between these to two pathways?

The results also point to Ficolin-1-rich granule membranes and Proteas some. How do these pathways may interact with energy metabolism making HFE pigs less feed efficient?

The overall conclusion is right on point but the explanation is completely lacking.

Response 19: Thank you very much for asking this question.

Based on proteomic screening and difference analysis we found that there were significant differences in these proteins between high and low groups and verified by PRM. Of course, these results are only based on correlation analysis and there is no causal mechanism research. The function of related proteins needs further experimental verification. These proteins are selected because these proteins are involved in the same signaling pathways at the same time. CXCL10 participates in TNF signaling pathway, CXCL10, STAT5B, RHOA, PRKCD, LSP1 were all significantly downregulated in the HFE group C-type lectin receptor signaling pathway, and Chemokine signaling pathway. These are immune-related signaling pathways.

This study has a certain experimental basis. In the early stage of our laboratory, we used the experimental samples (nasty 350) and the growth data (including feed efficiency, daily gain, average feed intake, etc.) to analyze the correlation between biochemical indexes and growth data. It was found that the correlation coefficient between TG and Glu in blood was the highest (r = 0.65, P < 0.0001). The second was the correlation coefficient between HDL-C and Glu, TG, and LDL-C (r = 0.51 to 0.59 P < 0.0001). TG was negatively correlated with body length and height, positively correlated with FCR, and corrected 30-100 kg FCR, and LDH was negatively correlated with FCR, corrected 30-100 kg FCR and RFI. SCr was negatively correlated with RFI, ADFI, FCR, corrected 30-100kg FCR, knot measured backfat thickness, and corrected 100kg backfat thickness (r = 0.13, r = 0.25, P < 0.05), and positively correlated with eye muscle area (r = 0.13, P < 0.05). CRP was significantly negatively correlated with eye muscle area measured by a knot, corrected eye muscle area by 100 kg, and corrected eye muscle thickness by 100 kg (r = 0.21, r = 0.23, P < 0.01), and positively correlated with FCR (r = 0.13, P < 0.01).

Glu and IL-6 were positively correlated with FCR and corrected 30-100 kg FCR (r = 0.12, 0.15, P < 0.05), and IL-6 was positively correlated with RFI and 100 kg backfat thickness EBV (r = 0.18, 0.21, P < 0.05). At the same time, the results of TNF- α, IL-6, and CRP in the H/LFE group were analyzed, and it was found that there was no significant difference in these three immune indexes between the two groups.

Therefore, there is a certain relationship between feed efficiency and immunity or inflammation. In this study, it is speculated that these proteins are potential candidates related to pig feed efficiency, and we will further study its mechanism.

Therefore, the conclusion of this paper is revised as the abundance of immune or inflammation-related proteins in the LFE group is higher, indicating that the immune efficiency of the HFE group may be lower than that of the LFE group.

Point 20:Ok, this is great. If the authors know they are closely related, so please try to explain how they work together to impact feed efficiency. If the authors believe it is through greater energy consumption by the immune system, so clearly describe the possible mechanism in the discussion.

Response 20:Thank you very much for your comments. We put forward this hypothesis and hope to further verify it in the follow-up experiments.

Point 21:It is not a conclusion.

Response 21:It has been modified in this article. The changes are as follows: In summary, the proteome of pig serum in HFE group and LFE group was compared and analyzed in this study

Point 22:It is true that activated immune system (e.g. when animals are ill) interfere with feed efficiency but considering that animals were healthy in the present study, the immune system was not activated. Therefore, the detected changes happened within basal immune status and, consequently, in the present study neither HFE animals have low immune status nor LFE have high immune status. This is the reason why, in a previous comment, this reviewer indicated that the authors should interpret these results the other way around, stating that high feed efficiency negatively impact the immune system.

Response 22:Thank you for your comments, which have been revised in the article.

Modified to: Therefore, HFE pigs have a less efficient immune system.

Point 23:This sentence is basically repeating the next one and can be removed.

Response 23:It has been modified in this article. Thank you!

Reviewer 2 Report

The authors performed proteomic analysis to determine the useful biomarker affecting to the feed efficiency phenotype in pigs’ plasma. They indicated that some inflammation-related pathways were significantly downregulated in high feed efficiency group. Their comparative data is valuable, and the analysis is well-constructed. The reviewer requested following modification.

1.       The author should state about statistical analysis in materials and methods section.

2.       In discussion section, they state the possibility about LSP1, PTPN11, CXCL10, STAT5B, RHOA, and PRKCD. However, the reviewer cannot follow about how the authors select these proteins.

3.       I think that protein lists in some important pathway in KEGG and gene ontology analysis should be present as Table.

4.       In gene ontology analysis, some pathways related to protein metabolism such as proteasome regulatory particles, ubiquitin- and modification-dependent protein catabolic processes, translation elongation factor activity etc. The protein metabolism pathway is related to feed efficiency?? Please discuss about this point.

5.       In the introduction, the previous omics approaches suggested glycolysis and mitochondrial metabolism is associated to porcine production. In authors’ proteomic data, the substrate-related to proteins are listed up?? Please state this point in discussion sections.

Author Response

Response to Reviewer 2 Comments

Point 1:The author should state about statistical analysis in materials and methods section.

Response 1:Dear reviewer, thank you very much for your kind advice. We have provided additional information in the Materials and Methods section, which is as follows :All growth data of experimental pigs were exported from the pig farm's comprehensive management information system (Kaifnets) and analyzed using SAS for data analysis and correlation testing. The RFI trait was obtained based on the calculation formula from Iowa State University, as follows:

RFI=ADFI-[b1i*(onBW-30)+b2i*(offBW-100)+b3imetamidBW+b4iADG1+b5i*BF].

Fisher's exact test was used to test for differentially expressed proteins, with a P-value <0.05 considered significant.

Point 2: In discussion section, they state the possibility about LSP1, PTPN11, CXCL10, STAT5B, RHOA, and PRKCD. However, the reviewer cannot follow about how the authors select these proteins.

Response 2:Thank you very much for asking this question. Based on proteomic screening and difference analysis we found that there were significant differences in these proteins between high and low groups and verified by PRM. Of course these results are only based on correlation analysis and there is no causal mechanism research. The function of related proteins needs further experimental verification. These proteins are selected because these proteins are involved in the same signaling pathways at the same time. CXCL10 participates in TNF signaling pathway,CXCL10, STAT5B, RHOA, PRKCD, Chemokine signaling pathway,LSP1, PTPN11, PRKCD and RHOA as well as C-type lectin receptor signaling pathway;. These are immune-related signaling pathways. This study has a certain experimental basis. In the early stage of our laboratory, we used the experimental samples (nasty 350) and the growth data (including feed efficiency, daily gain, average feed intake, etc.) to analyze the correlation between biochemical indexes and growth data. It was found that the correlation coefficient between TG and Glu in blood was the highest (r = 0.65, P < 0.0001). The second was the correlation coefficient between HDL-C and Glu, TG and LDL-C (r = 0.51 to 0.59 P < 0.0001). TG was negatively correlated with body length and height, positively correlated with FCR and corrected 30-100 kg FCR, and LDH was negatively correlated with FCR, corrected 30-100 kg FCR and RFI. SCr was negatively correlated with RFI, ADFI, FCR, corrected 30-100kg FCR, knot measured backfat thickness and corrected 100kg backfat thickness (r = 0.13, r = 0.25, P < 0.05), and positively correlated with eye muscle area (r = 0.13, P < 0.05). CRP was significantly negatively correlated with eye muscle area measured by knot, corrected eye muscle area by 100 kg and corrected eye muscle thickness by 100 kg (r = 0.21, r = 0.23, P < 0.01), and positively correlated with FCR (r = 0.13, P < 0.01). Glu and IL-6 were positively correlated with FCR and corrected 30-100 kg FCR (r = 0.12, 0.15, P < 0.05), and IL-6 was positively correlated with RFI and 100 kg backfat thickness EBV (r = 0.18, 0.21, P < 0.05). At the same time, the results of TNF- α, IL-6 and CRP in H/LFE group were analyzed, and it was found that there was no significant difference in these three immune indexes between the two groups. Therefore, there is a certain relationship between feed efficiency and immunity or inflammation. In this study, it is speculated that these proteins are potential candidates related to pig feed efficiency, and we will further study its mechanism. Therefore, the conclusion of this paper is revised as that the abundance of immune or inflammation-related proteins in LFE group is higher, indicating that the immune efficiency of HFE group may be lower than that of LFE group.

Point 3: I think that protein lists in some important pathway in KEGG and gene ontology analysis should be present as Table.

Response 3:Thank you very much for the suggestion you have made. I have made modifications to the manuscript that I uploaded earlier, and uploaded the results of KEGG and GO enrichment analyses to Attachment 4.

Point 4: In gene ontology analysis, some pathways related to protein metabolism such as proteasome regulatory particles, ubiquitin- and modification-dependent protein catabolic processes, translation elongation factor activity etc. The protein metabolism pathway is related to feed efficiency?? Please discuss about this point.

Response 4:We have added relevant explanations in the third paragraph of the discussion. The content of the explanations is as follows: There is a close relationship between protein metabolism pathway and feed efficiency in animals. Protein metabolism pathway refers to the process of digestion, absorption, transport, synthesis, and degradation of protein in animals, while feed efficiency refers to the ratio of productivity or weight gain to feed intake. The impact of protein metabolism pathway on feed efficiency can be summarized as follows:

  • Digestion and absorption efficiency: factors such as digestive enzyme activity, intestinal health status, and amino acid supply can affect the digestion and absorption efficiency of protein, and thus affect feed utilization and feed efficiency.
  • Protein synthesis: protein synthesis requires multiple amino acids and other nutrients. If the content of certain essential amino acids in the feed is insufficient, it will limit protein synthesis and productivity, thereby reducing feed efficiency.
  • Physiological state: animal's demand for protein and its utilization efficiency may vary under different physiological conditions, which also affects feed efficiency.

Therefore, understanding the protein metabolism pathway is crucial for optimizing feed formulation and improving feed efficiency.

Point 5: In the introduction, the previous omics approaches suggested glycolysis and mitochondrial metabolism is associated to porcine production. In authors’ proteomic data, the substrate-related to proteins are listed up?? Please state this point in discussion sections.

Response 5 Thank you very much for raising this question. In this experiment, differential expression proteins were found to be enriched in the glycoprotein and mitochondrial metabolism signaling pathways through GO and KEGG analysis. Among them, PKM protein was enriched in the glycolysis signaling pathway, and it was found that different structural forms of PKM protein were significantly upregulated or downregulated in both LFE and HFE groups. I believe that further research is needed to understand the function of PKM, so it was not discussed in the paper. SLC25A13, FBXL4, and ALAS2 proteins were all found to be enriched in the mitochondrial metabolism pathway. SLC25A13 protein is a transport protein located on the inner membrane of mitochondria, also known as citrate/malate transporter (CIC). It mainly participates in the TCA cycle and fatty acid metabolism in the human body. Although current research is limited, it has been shown that the loss or mutation of SLC25A13 protein can reduce the activity of AMPK, leading to cellular energy metabolism disorders and mitochondrial dysfunction. FBXL4 protein is mainly a subunit of mitochondrial respiratory chain complex V, which participates in regulating the internal oxidative phosphorylation process of mitochondria. Research on its signal pathway is still limited, but it has been shown that the abnormal function of FBXL4 protein may involve multiple metabolic pathways and signaling pathways. For example, the loss or mutation of FBXL4 protein can affect the normal operation of the internal fatty acid metabolism pathway of mitochondria, leading to metabolic disorders and various metabolic diseases. Recent studies have also found that FBXL4 protein is related to the mTORC1 (mammalian target of rapamycin complex 1) signaling pathway. FBXL4 protein may be involved in the negative regulation of the mTORC1 signaling pathway, thus affecting the metabolic balance inside cells. Because the above two proteins were not quantified during PRM validation, no relevant discussion was included in this paper. However, since these proteins have been found to be enriched, I believe that further research can be conducted. ALAS2 is the first rate-limiting enzyme in heme biosynthesis pathway. If ALAS2 undergoes mutation or inhibition, it may lead to heme synthesis disorders, which is not related to the purpose of this study and hence was not discussed.

Round 2

Reviewer 1 Report

Dear authors,

The quality of the images was improved but it would be advisable to improve them even further. The discussion was not very much modified which would leave the explanation of the results disapointing but by the end of the discussion the authors clearly stated that "it is speculated that these proteins are potential candidates", meaning that they are not sure about it and that they "will further study the mechanisms". Maybe such statement should be moved to the beginning of the discussion.

I believe it will guide the readers in the understanding that much is needed to get into conclusions about the reliability of these markers anf their eventual mechanisms of action.

Thank you

Author Response

Response to Reviewer Comments

Point: The quality of the images was improved but it would be advisable to improve them even further. The discussion was not very much modified which would leave the explanation of the results disapointing but by the end of the discussion the authors clearly stated that "it is speculated that these proteins are potential candidates", meaning that they are not sure about it and that they "will further study the mechanisms". Maybe such statement should be moved to the beginning of the discussion.

Response: Thank you for your excellent suggestions and the changes I have discussed in this submission. I redrew the unclear pictures in the article, replaced the unclear images with higher resolution, and submitted the clear pictures to the editor. In the revision of the discussion part, it is first pointed out that the proteins discussed next are related to porcine immune inflammation and are discussed as potential candidates for FE. Through the existing research reports, this paper discusses the function of the protein and expounds on the relationship between immunity and feed efficiency, which increases the rationality of this paper.